# Hierarchical true prevalence, risk factors and clinical symptoms of tuberculosis among suspects in Bangladesh

**Mohammad Kamruzzaman Khan**[1,2], **Md. Nazimul Islam**[1], **Jayedul Hassan**[3], **Shaymal Kumar Paul**[4], **M. Ariful Islam**[1], **Konstantinos Pateras**[5], **Polychronis Kostoulas**[5], **Michael P. Ward**[6], **A. K. M. Anisur Rahman**[1], **Md. Mahbub Alam**[1] *

1 Faculty of Veterinary Science, Department of Medicine, Bangladesh Agricultural University, Mymensingh, Bangladesh, 2 Department of Community Medicine, Mymensingh Medical College, Mymensingh, Bangladesh, 3 Faculty of Veterinary Science, Department of Microbiology and Hygiene, Bangladesh Agricultural University, Mymensingh, Bangladesh, 4 Department of Microbiology, Netrokona Medical College, Netrokona, Bangladesh, 5 Laboratory of Epidemiology & Artificial Intelligence, Faculty of Public Health, University of Thessaly, Volos, Greece, 6 Sydney School of Veterinary Science, The University of Sydney, Camden, New South Wales, Australia

* asamahbub2003@yahoo.com

**Data Availability Statement:** All relevant data are within the paper and its Supporting Information files.

## Abstract

### Background

The study was aimed to estimate the true prevalence of human tuberculosis (TB); identify risk factors and clinical symptoms of TB; and detect rifampicin (RIF) sensitivity in three study areas of Bangladesh.

### Methods

The cross-sectional study was conducted in three Bangladesh districts during 2018. Potential risk factors, clinical symptoms, and comorbidities were collected from 684 TB suspects. Sputum specimens were examined by LED microscopy. TB hierarchical true prevalence, risk factors and clinical symptoms were estimated and identified using a Bayesian analysis framework. Rifampicin sensitivity of *M. tuberculosis* (MTB) was detected by GeneXpert MTB/RIF assay.

### Results

The median TB true prevalence was 14.2% (3.8; 34.5). Although overall clustering of prevalence was not found, several DOTS centers were identified with high prevalence (22.3% to 43.7%). Risk factors for TB identified (odds ratio) were age (> 25 to 45 years 2.67 (1.09; 6.99), > 45 to 60 years 3.43 (1.38; 9.19) and individuals in families/neighborhoods where a TB patient(s) has (ve) already been present (12.31 (6.79; 22.60)). Fatigue, night sweat, fever and hemoptysis were identified as important clinical symptoms. Seven of the GeneXpert MTB/RIF positive sputum specimens (65) were resistant to rifampicin.

**Funding:** This research work was funded by the United States Department of Agriculture (USDA) [Project/Grant ID: USDA2001 Section 416 (b)]. The funders had no role in study design, data collection and analysis, decision to publish, or preparation of the manuscript.

**Competing interests:** The authors have declared that no competing interests exist.

## Conclusions

About one in every seven TB suspects was affected with TB. A number of the TB patients carry multi drug resistant MTB. Hierarchical true prevalence estimation allowed identifying DOTS centers with high TB burden. Insights from this study will enable more efficient use of DOTScenters-based TB surveillance to end the TB epidemic in Bangladesh by 2035.

## Introduction

TB has accompanied mankind during its evolution [1] and continues to exert an enormous toll on human health despite the availability of effective anti-tubercular drugs for more than 50 years. There are about 10.4 million new *Mycobacterium tuberculosis* infections and 1.4 million deaths per year and TB is one of the top 10 causes of death worldwide [2]. Estimated global TB incidence and global TB mortality, in 2017, was 133 and 21 per 100 000 populations, respectively [3]. The global distribution of TB cases is overrepresented by low-income countries with emerging economies. The highest prevalence of cases is in Asia where China, India, Bangladesh, Indonesia and Pakistan collectively make up over 50% of the global burden [4]. TB is often described as a barometer of social welfare. Poor quality of life, poor housing, overcrowding, under-nutrition, smoking, lack of education, large families, lack of awareness regarding cause and spread of TB– all these social factors are interrelated and contribute to the occurrence and transmission of TB [5].

Clinical manifestation varies with TB type and immune status of the patient. Pulmonary complaints are evident in 85% of cases and clinical features significantly associated with active pulmonary TB are cough with expectoration for 2 weeks, low grade evening fever, hemoptysis, loss of appetite, weight loss, fatigue, chest pain and shortness of breath are common [6–8]. Diagnosis of pulmonary TB is difficult and often requires a combination of tests. Smear microscopy is rapid but with a poor sensitivity which rarely exceeds 68% [9]. Culture-based methodology–in solid or liquid media–is considered the gold standard, but it requires several weeks to months for a result [10]. Emergence of multidrug resistant TB (MDR TB) is another limitation in TB diagnosis and control as it requires sophisticated diagnostic methods and hampers treatment success [11]. To overcome these limitations, rapid screening methods such as GenoTypeMTBDR*plus* line probe assay (LPA) and GeneXpert MTB/RIF were introduced [12]. These methods allow rapid detection of *M. tuberculosis* along with resistance against rifampicin and/or isoniazid, which are commonly used to treat TB [12].

Bangladesh is one of the countries with the highest TB burden worldwide [13]). The country has been implementing a National Tuberculosis Control Program (NTP) since 1965.A project based on the DOTS (Directly Observed Treatment, Short course) strategy was initiated in Bangladesh in 1993 [14]. Anyone willing to test for TB can attend a DOTS center free of cost. However, health workers visit every household monthly and recommend people having cough for >2 weeks to visit DOTS center for TB testing. The DOTS strategy is now being implemented throughout the whole country via 1147 DOTS centers which have sputum smear microscopy facilities [https://www.ntp.gov.bd/DOTS list]. The DOTS centers are a one stop service where patients receive free diagnostic facilities and supervised treatment with anti-TB drugs. The DOTS strategy helps to reduce treatment failure, loss to follow up and development of MDR TB.

Previous studies aiming to elucidate aspects of the human TB epidemiology in Bangladesh were based on the general population or rural areas or slum dwellers [6,7,15]. DOTS centers

have been widespread throughout the country since 2007 (100% DOTS coverage) and people are also aware about free diagnosis and treatment. DOTS centers have successfully detected and treated 96.4% of smear microscopy confirmed new pulmonary TB cases in 2018 [16]. DOTS centers are dedicated for the surveillance and control of TB in Bangladesh. However, the true prevalence of TB, identification of risk factors and clinical symptoms among TB suspects attending DOTS centers have not been studied previously. Therefore, the aims of this study were to (i) estimate true prevalence, ii) identify risk factors and clinical features of TB and (ii) detect rifampicin sensitivity among TB suspects in Bangladesh.

## Methods

### Ethical approval

The study was approved by the Institutional Review Board (IRB) of Mymensingh Medical College, Mymensingh, Bangladesh (Memo no. MMC/IRB/142, date: 18/11/2017). Written informed consent was obtained from all the participants before recruitment. Records were kept strictly confidential.

### Design, location and duration of study

The cross-sectional study was conducted from January to December 2018 among the TB suspects attending DOTS centers in the Mymensingh, Sirajganj and Dhaka districts of Bangladesh.

### Sample size calculation and sampling techniques

Using the Guilford and Frucher formula ($n = \frac{z^2 pq}{d^2}$) [17], assuming a prevalence of 50% [p = 0.5] (no previous report) among the subpopulation suspect of TB in Bangladesh, the minimum calculated sample size was 600, at a significance level of 5% and with 4% acceptable margin of error. From 8 administrative divisions of Bangladesh we selected 3 divisions randomly. Again, from each selected division we selected one district randomly namely Mymensingh, Sirajganj and Dhaka. From each district 8 DOTS centers were selected randomly i.e. ultimately, we randomly selected 24 DOTS centers from Mymensingh, Sirajganj and Dhaka districts. Finally, a convenience sample of 684 TB suspects attending those DOTS centers were interviewed using pretested, semi-structured case record forms, which included information on symptoms, socio-demographic characteristics and risk factors for TB.

### Collection of specimens

Each TB suspect brought a morning sputum specimen in the plastic container provided by the NTP. After conducting face to face interview another sputum specimen from each TB suspect was collected on the spot.

### Auramine staining and LED microscopy

Thin smear slides were made from the purulent part of the sputum. Auramine-rhodamine was used as a fluorescent dye to stain acid-fast bacteria (AFB) and examined under LED microscope on high power (400X). Acid-fast organisms fluoresced bright yellow or orange against a dark background [18].

## GeneXpert MTB/RIF assay

GeneXpertMTB/RIF assay was performed following the protocol of the manufacturer (Cepheid Inc., Sunnyvale, CA, USA). Sputum specimens were collected in containers provided and treated with sample reagent in a proportion of 2:1 and incubated for 15 minutes at room temperature. Two milliliters reagent treated sample was pipetted into the sample chamber of the Xpert cartridge. The Xpert cartridge was then placed into the GeneXpert instrument system and run. Results were generated after 90 min.

## Statistical analysis

An individual was considered to be TB infected if he/she was positive by LED microscopy after auramine staining. TB data were entered into a spreadsheet (Microsoft Excel 2010) and transferred to R 4.0.2 [19] for analysis. Age was converted to a categorical variable based on quartiles. The frequency and proportion of TB in each category of independent variables were calculated using the "tabpct" function of the R package "epiDisplay" [20]. The data used for the identification of clinical symptoms and risk factors is available as a S1 File.

**True prevalence.**   The overall and DOTS center level true prevalence of tuberculosis among TB suspects in Bangladesh were estimated using a shiny web-application tPRiors [21]. The sensitivity (mean = 0.627 and lower limit at 95% confidence level = 0.50) and specificity (mean = 0.987 and lower limit at 95% confidence level = 0.90) of the auramine staining and LED microscopy [22]. was used as prior information in the model. This prior information produced Beta (18.34, 10.77) for sensitivity and Beta (95.32, 5.02) for specificity of auramine staining and LED microscopy in the Bayesian model. As there was no previous report on the prevalence of human TB among TB suspects in Bangladesh we used a diffused prior for the prevalence [Beta(0.8292, 0.8246)]. For this Beta distribution we used the following prior information in the tPRiors: 50% mean prevalence, 95% level of confidence that the true value of the mean is greater than the percentile value, the upper limit of the mean as 80.0% at 95% level of confidence, the level of confidence that a certain fraction of the units under study has a prevalence less than the 'percentile.median' as 81.1%, the median value of the defined 'psi.percentile' as 82% and the value that the 'percentile.median' does not exceed 90% with 95% confidence. The influence of prior information on the posterior true prevalence was evaluated by providing no prior information in the model [Beta (1,1)]. The model was run using three Markov chains with 100,000 iterations each, half of which were discarded as burn in. The data used for the estimation of hierarchical true prevalence of human TB is provided as a S2 File.

**Mixed-effects Bayesian logistic regression–Univariable screening.**   To assess the association between the individual TB status and potential risk factors we used mixed-effect univariable logistic regression models with DOTS center as a random-effects term, within a Bayesian estimation framework. All candidate explanatory variables were initially screened, one-by-one. Variables with a Bayesian p-value <0.25 were then offered to a full model which was, subsequently reduced by backwards elimination, until only significant (P<0.05) variables remained. Collinearity among explanatory variables was assessed by Cramer's phi-prime statistic (R package "vcd," "assocstats" function). A pair of variables was considered collinear if Cramer's phi-prime statistic was >0.70 [23].

**Multivariable mixed-effects Bayesian logistic regression.**   Two separate Bayesian logistic regression models with a random-effects term at the DOTS center level were used to identify (i) risk factors for and (ii) clinical symptoms associated with TB according to the previously described method [24]. Confounding was checked by observing the change in the estimated coefficients of the variables that remained in the final model by adding a non-selected variable to the model. If the inclusion of this non-significant variable led to a change of more than 25%

of any parameter estimate, that variable was considered a confounder and retained in the model [25]. The two-way interactions of all variables remaining in the final model were assessed for significance based on AIC values, rather than significance of individual interaction coefficients. The spatial autocorrelation of TB prevalence was calculated by the "Moran.I" command of the R package "ape" [26], using sub district centroids. Likewise, spatial autocorrelation of residuals of the final logistic regression model was estimated.

### Statistical software

Bayesian logistic regression models were run using the "stan_glmer" function of the R package "stanarm" [27]. The model was run using five Markov chains with 16000 iterations each, half of which were discarded as burn in. The models' convergence, diagnostics, posterior estimates and posterior predictive checks were observed via a user-friendly graphical user interface of the "launch_shinystan" function of the "shinystan" R package [28]. The R code used for the Bayesian mixed-effects logistic regression analysis with step-by-step explanation is provided in S3 File. The analysis was performed in R 4.0.2 [19] and tPRiors [21].

## Results

### Descriptive results

Sputum from 684 TB suspects attending 24 DOTS centers in Mymensingh, Sirajganj and Dhaka districts were examined under LED microscope after auramine staining and 80 (11.7%; 95% Confidence Interval: 9.4; 14.4) were found positive for tubercle bacilli (Table 1). About 60% (50.3; 69.3) of the tested (108) individuals were GeneXpert MTB/RIF assay positive. Among them 89.2% (78.5; 95.2) and 10.8% (4.8; 21.5) were rifampicin sensitive and resistant, respectively. Child TB was observed only among (3/32) 9.4% (2.5; 26.2) children tested.

### True prevalence

The overall median true prevalence of tuberculosis among TB suspects was 14.2% (3.8; 34.5) (Fig 1, S4 File). Tuberculosis was predominantly detected in Fulpur (43.7%: 0.2; 87.2) followed by Trishal (43.3%: 16.4; 70.3), Nandail (26.2%: 16.9; 69.3), Gauripur (14.7%: 9.8; 39.2), Fulbaria (22.3%: 1.2; 43.3), Sirajganj Sadar (18.9%: 6.7; 30.9) and Muktagacha DOTS centers (17.7%: 3.8; 31.6). The model without any prior information also yielded the median prevalence of 14.1% (1.4; 32.3) indicating robustness of the model to estimate the posterior TB true prevalence. The spatial distribution of TB prevalence did not show significant clustering (Moran's I = 0.0171, P = 0.3185).

### Risk factors for TB

Age and presence of tuberculosis patient in the family or neighborhood were identified as risk factors for TB. The odds of TB were 2.67 (1.09; 6.99) and 3.43 (1.38; 9.19) times higher among

**Table 1. Results of different tests for the diagnosis of human tuberculosis (n = 684) in selected districts of Bangladesh, 2018.**

| Tests | Tested | Negative | Positive (%) | 95% CI |
|---|---|---|---|---|
| Auramine staining and LED microscopy | 684 | 504 | 80 (11.7) | 9.4; 14.4 |
| GeneXpert MTB/RIF | 108* | 43 | 65 (60.2) | 50.3; 69.3 |
| | | | Sensitive:58 (89.2) | 78.5; 95.2 |
| | | | Resistant: 7 (10.8) | 4.8; 21.5 |

CI: Confidence Interval,*108 samples out of 684 were tested by GeneXpert MTB/RIF.

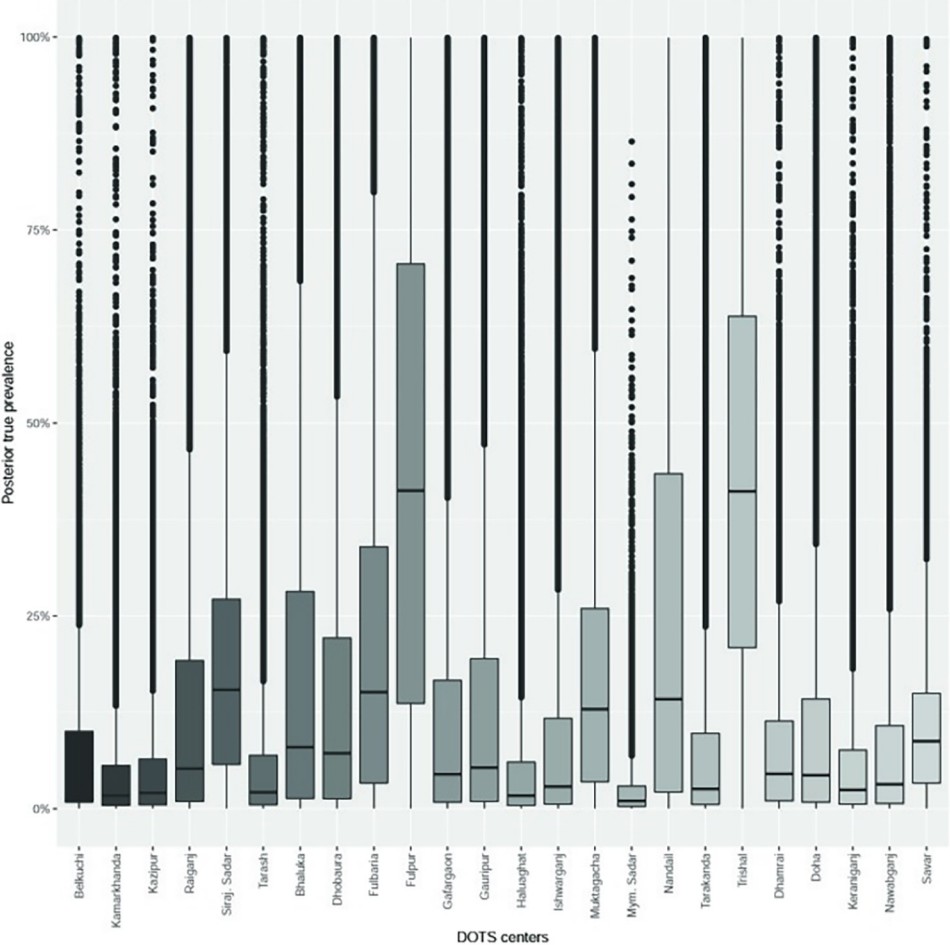

**Fig 1. Boxplots showing the DOTS centers level true prevalence of TB among suspects in Mymensingh, Sirajganj and Dhaka districts of Bangladesh.**

TB suspects aged > 25 to 45 years and > 45 to 60 years, respectively, than those aged > 60 years. Moreover, the presence of TB patient in the family or neighborhood increased the risk of TB by 12.31 (6.79; 22.60) times compared to families or neighborhoods without TB patients (Table 2). Results of the univariable pre-screening are provided as a S5 File. Multicollinearity was not detected among the selected explanatory variables for multivariable regression.

## Clinical symptoms and comorbidities associated with TB

Individuals with fatigue and experiencing night sweats were 4.8 (2.6; 9.0) and 8.7 (2.5; 31.9) times more likely to be TB positive, respectively. Individuals with hemoptysis and fever had 9.6 (3.7; 25.8) and 9.1 (2.6; 43.3) times higher odds of being TB positive, respectively (Table 3). Results of the univariable pre-screening are provided as a S6 File. No multicollinearity was detected among the selected explanatory variables for multivariable regression.

Both risk factor and clinical symptoms models converged well (the R-hat values were less than 1.1) and the effective sample sizes were more than the total number of iterations (Tables 2 and 3). The trace plots exhibited good mixing and showed no signs of convergence problems. The graphical posterior predictive check revealed that our models adequately fit the data. The

**Table 2. Risk factors retained in the final Bayesian mixed-effects multivariable logistic regression model for tuberculosis (TB) among TB suspects in Mymensingh, Sirajganj and Dhaka, Bangladesh.**

| Risk factors | Category | Estimate | SD | Odds ratio (95% CrI) | n_eff |
|---|---|---|---|---|---|
| Age (years) | ≤ 25 | 0.58 | 0.52 | 1.79 (0.66, 5.20) | 23,750 |
| | >25 to 45 | 0.97 | 0.47 | 2.67 (1.09, 6.99) | 22,147 |
| | > 45 to 60 | 1.23 | 0.48 | 3.43 (1.38, 9.19) | 22,554 |
| | > 60 | - | - | Reference | |
| Tuberculosis patient in the family or neighborhood | Yes | 2.51 | 0.30 | 12.31(6.79, 22.60) | 43,510 |
| | No | - | - | Reference | |

SD = Standard deviation, CrI: Credible Interval, n_eff = effective sample size, RHat is 1 for all parameters.

residuals of the risk factor model did not show significant spatial autocorrelation (Moran's I = -0.0025, P = 0.5673).

## Discussion

This study was conducted on 684 TB suspects visiting DOTS centers in Mymensingh, Sirajganj and Dhaka districts of Bangladesh. Overall, one in every seven TB suspects was found to be affected by TB and some DOTS centers had a high TB burden. We identified risk factors for TB and clinical symptoms associated with TB positivity. About 11% multidrug resistant MTB (MDR TB) was detected in the study areas. The true prevalence and MDR TB were high among TB suspects. The results of this study will enhance existing TB surveillance by targeting DOTS centers with high prevalence and people with risk factors and clinical symptoms and allocating resources for the management of TB and MDR TB.

Age and presence of a TB patient at home or neighborhood were risk factors for TB. The risk of developing TB is high in close family members compared to more distant relatives [29]. A small number of families with micro-epidemics are responsible for most of the new TB cases which are more infectious. There is an extremely high risk of transmission of TB in these families [30]. Socio-demographic characteristics of the patients, such as age, influences the occurrence of tuberculosis [7,15]. The economically most active age group (15–54 years) is more vulnerable to TB than the inactive or older age group [8].

Smoking is a well known risk factor for pulmonary tuberculosis in humans [31,32]. We did not find a significant association between smoking and TB in this study. Also, smoking was

**Table 3. Clinical symptoms significantly associated with human tuberculosis in the Bayesian mixed-effects multivariable logistic regression model.**

| Clinical symptoms | Category | Estimate | SD | Odds ratio (95% CrI) | n_eff |
|---|---|---|---|---|---|
| Fatigue | Yes | 1.57 | 0.32 | 4.8 (2.6; 9.0) | 56,587 |
| | No | - | - | Reference | |
| Night sweat | Yes | 2.17 | 0.64 | 8.7 (2.5; 31.9) | 54,441 |
| | No | - | - | Reference | |
| Hemoptysis | Yes | 2.26 | 0.49 | 9.6 (3.7; 25.8) | 50,872 |
| | No | - | - | Reference | |
| Fever | Yes | 2.21 | 0.72 | 9.1 (2.6; 43.3) | 38,788 |
| | No | | | | |

SD = Standard deviation, CrI = Credible Interval, n_eff = effective sample size, RHat = 1 for all parameters.

not identified as a confounder of the relationship between TB and age and presence of a TB patient at home or neighborhood in this study (S7 File).

Evidence of transmission of TB from animals to humans, which has been reported and could be of major concern in Bangladesh, was out of the scope of this study. Also, a range of potential risk factors such as drinking raw milk, taking care of cattle, handling raw milk or meat, and presence of cattle in the family (S5 File) were not associated with TB occurrence. Nevertheless, transmission cannot be excluded because cattle were not tested for TB and potential molecular associations were not investigated. TB transmission from cattle to humans has been reported through close contact or handling of cattle milk or meat and drinking raw milk or consumption of undercooked meat or lack of protective measures during slaughtering of cattle [4,33].

We identified fatigue, night sweats, hemoptysis, and fever (but not the presence of a long lasting cough) as clinical symptoms associated with TB occurrence. Hence, patients with fatigue and/or night sweats and/or hemoptysis and/or fever should be rigorously examined for TB even in the absence of coughing. Fever, cough, fatigue, weight loss, chest pain, hemoptysis, difficult breathing, night sweats, and loss of appetite are the common clinical symptoms associated with TB [6,7].

After adjusting for the sensitivity and specificity of smear microscopy the median true prevalence of TB among tested people was 14.2%. Previous tuberculosis prevalence studies in Bangladesh were not DOTS center based and hence we used diffused prior information in the model to give more weight to the data [23]. However, even if we use 20% prior prevalence in the model the posterior median prevalence does not change much [14.9%] (S8 File). We identified several DOTS centers with high TB prevalence including Fulpur, Trishal, Nanadail, Muktagacha, Fulbaria and Sirajganj Sadar. Although overall no significant clustering was detected, two subdistricts (Fulpur and Trishal) had higher prevalence (43.7% and 43.3%). The autocorrelation of the residuals of the risk factor model was also non-significant, indicating that this model adequately explained the spatial distribution of risk. The existing TB surveillance should prioritize DOTS centers with high prevalence for more efficient TB control programs. The true prevalence of human TB in different DOTS centers will also enable policy planners to allocate resources for TB treatment. Further investigation of subdistricts with high TB prevalence is warranted, with a focus on MDR TB. We included DOTS centers in 3 divisions out of eight and the health service has a workforce to reach every household in Bangladesh to convey health related information. DOTS centers are uniformly located throughout the country including cities, urban and rural areas, which include people of all sociodemographic conditions. So, these prevalence and risk factors estimates likely represent the whole country.

Almost 11% of the GeneXpert MTB/RIF positive cases were infected with multidrug resistant TB in the study areas. The reported incidence of MDR/RR TB was 3.7 per 100,000 populations in 2018 and 2.0 per 100,000 populations in 2019 in Bangladesh [3,13]. We estimated MDR TB among TB patients, whereas the two aforementioned incidence reports were based on the general population [i.e. the denominators are different]. That might be the possible reason for the higher MDR TB prevalence in this study. This finding will also enable policy planners to allocate resources for the treatment of multidrug resistant TB. The WHO-endorsed DOTS Plus strategy that adds components for MDR TB diagnosis, management and treatment, should be implemented in these areas with high MDR TB burden.

TB diagnosis in Bangladesh is mostly dependent on microscopy because of its low cost. However, the sensitivity of this method is about 50% [34]. High-tech diagnostics such as GeneXpert are very expensive but efficient (sensitivity 93%) to detect TB [35]. Although GeneXpert

would give a better estimate of drug resistant *Mycobacterium tuberculosis* in the study population, we could not screen all samples due to financial constraints.

## Conclusion

About one in every seven TB suspects was affected with TB and some DOTS centers have high TB burden. We identified age and neighboring infections as risk factors for TB, and fatigue and/or night sweats and/or hemoptysis and/or fever but not coughing as symptoms that should guide TB testing. Around 11% of the TB patients carry multi drug resistant MTB in the study areas. These results can contribute to the more efficient use of DOTS center-specific surveillance and risk-based NTC program.

## Supporting information

**S1 File. Data used to identify clinical symptoms and risk factors for human tuberculosis among humans attending at DOTS centers.**
(XLSX)

**S2 File. Data used to estimate the true prevalence of tuberculosis among humans attending at DOTS centers.**
(XLS)

**S3 File. The R code used to identify clinical symptoms and risk factors for human tuberculosis using Bayesian mixed-effects logistic regression models.**
(TXT)

**S4 File. tPRiors dynamic report of the main Bayesian model to estimate the overall and DOTS centers based true prevalence of human TB.**
(PDF)

**S5 File. Univariable association of demographic and other risk factors with human tuberculosis (TB) in Mymensingh, Sirajganj and Dhaka, Bangladesh (n = 684).**
(DOCX)

**S6 File. Clinical symptoms and comorbidities associated with human tuberculosis (TB) based on univariable mixed-effects Bayesian logistic regression analyses (n = 684).**
(DOCX)

**S7 File. Changes in the odds ratio after adding smoking in the final multivariable model.**
(DOCX)

**S8 File. The influence of informative prior on the posterior overall and DOTS centers based human TB true prevalence of in Bangladesh.**
(PDF)

## Acknowledgments

The authors express thanks to Damien Foundation, Mymensingh for technical assistance in culturing sputum specimens.

## Author Contributions

**Conceptualization:** M. Ariful Islam, Md. Mahbub Alam.

**Data curation:** Mohammad Kamruzzaman Khan.

**Formal analysis:** Konstantinos Pateras, Polychronis Kostoulas, Michael P. Ward.

**Funding acquisition:** M. Ariful Islam, Md. Mahbub Alam.

**Investigation:** Mohammad Kamruzzaman Khan, Md. Nazimul Islam, Shaymal Kumar Paul.

**Methodology:** Mohammad Kamruzzaman Khan, Md. Nazimul Islam, Jayedul Hassan, Shaymal Kumar Paul, Michael P. Ward, A. K. M. Anisur Rahman.

**Software:** Konstantinos Pateras, Polychronis Kostoulas, A. K. M. Anisur Rahman.

**Supervision:** Md. Mahbub Alam.

**Writing – original draft:** Mohammad Kamruzzaman Khan, Md. Nazimul Islam, Jayedul Hassan.

**Writing – review & editing:** Jayedul Hassan, Shaymal Kumar Paul, M. Ariful Islam, Konstantinos Pateras, Polychronis Kostoulas, Michael P. Ward, A. K. M. Anisur Rahman, Md. Mahbub Alam.

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
