## [Decision Letter · Decision Letter 0]

28 Apr 2022

PONE-D-22-00718Hierarchical true prevalence, risk factors and clinical symptoms of tuberculosis among suspects in BangladeshPLOS ONE

Dear Dr. Alam,

Thank you for submitting your manuscript to PLOS ONE. After careful consideration, we feel that it has merit but does not fully meet PLOS ONE’s publication criteria as it currently stands. Therefore, we invite you to submit a revised version of the manuscript that addresses the points raised during the review process.

The manuscript is well written, however the comments from reviewer 3 about specificity needs to be adequately addressed in the manuscript.Please ensure that your decision is justified on PLOS ONE’s publication criteria and not, for example, on novelty or perceived impact.

We look forward to receiving your revised manuscript.

Kind regards,

Pradeep Kumar, Ph.D.

Academic Editor

PLOS ONE

Journal Requirements:

Reviewers' comments:

Reviewer's Responses to Questions

**Comments to the Author**

1. Is the manuscript technically sound, and do the data support the conclusions?

Reviewer #1: Yes

Reviewer #2: Yes

Reviewer #3: Partly

2. Has the statistical analysis been performed appropriately and rigorously? 

Reviewer #1: I Don't Know

Reviewer #2: Yes

Reviewer #3: Yes

3. Have the authors made all data underlying the findings in their manuscript fully available?

Reviewer #1: Yes

Reviewer #2: Yes

Reviewer #3: No

4. Is the manuscript presented in an intelligible fashion and written in standard English?

Reviewer #1: Yes

Reviewer #2: Yes

Reviewer #3: Yes

5. Review Comments to the Author

Reviewer #1: This Manuscript is technically sound and easy to understand with enough supporting data. However, this type of studies was carried out previously in Bangladesh. It was not clear why DOTS data is critical in compare to the previous studies. DOTS centers have been widespread since 2007, authors did not identified/clarified about whether sample population from previous studies visited DOT center or not. Furthermore, authors highlighted MDR MTB in the conclusion with limited number of GeneXPert data due to financial limitation. It is suggested not to highlight the percentage of the MDR in the conclusion.

Reviewer #2: The authors in this manuscript investigated the true prevalence of TB among TB suspects and identified their risk factors and clinical symptoms. They also studied the rifampin sensitivity in those selected populations. The entire study was conducted in Bangladesh which has high prevalence of TB. The manuscript is well written and the findings described here would be of interest for better surveillance management of TB surveillance.

LIne 221-227; 296-298: What is the socio-economic difference between Fulpur and Muktagacha DOTS centers? Was there any previous report/observation of high and detection level between these two sites?

Line 308: Is there any other studies on TB determining the true prevalence in other parts of the world? Moreover, based of socio-economic difference of the major cities, urban and rural areas, how did the authors claim that TB suspects can be representative of the whole country?

Line 309: Is there any previous report of Rif resistant TB in Bangladesh? If yes, please provide reference and compare your findings.

Line 315: Please provide journal reference for microscopy dependent TB detection sensitivity.

Reviewer #3: The manuscript entitled “Hierarchical true prevalence, risk factors and clinical symptoms of tuberculosis among suspects in Bangladesh” has been read carefully. Authors here would like to emphasize on the scenario of true prevalence of tuberculosis around suspected areas of Bangladesh. Their aim to provide information on true prevalence and risk factors contributing to TB transmission is encouraging. I would like to provide my concerns listed below which could enhance their manuscript for acceptance:

1) The foremost concern of this study is their assumption for TB detection based on a less sensitive cum specific technique called auramine staining and they mentioned it that its not possible to test all samples by Xpert method. I understand this limitation, but specificity is a major concern from sputum by using Auramine staining. NTM’s could also get detected leading to misdiagnose if the staining is not supported with some better culture techniques for a greater number of samples.

2) I would not like the statement (line 26 in abstract) in Bangladesh until there are more subjects under investigation. Using province or some other word would better justify their mini study.

3) There are only two criteria to place suspected individuals in TB category one is sputum smear which is a poor technique. Saying that culture takes longer time for results but is a gold standard is not enough, since this study started 2 years back, authors may have gotten enough time to screen samples through culture to better understand this criterion.

4) Are there any false positives in their study design with the staining process, considering the specificity is only 68 %?

5) In selecting their divisions from specific areas of Bangladesh, they use random process, do they mean there is no prior information about number of people visiting, having likely TB symptoms?

6) Similarly, for concluding risk factors for mixed effects multivariable logistics regression model, is there any prior value for the estimation of ODDS?

7) The Odds ratios are impressive for the risk factors in univariable and multivariable logistic regression Bayesian model, I would like to know why only two risk factors are retained in the final model, is it because of the insignificant odds? It would have been interesting to understand how inclusion of smoking would affect the odds.

8) Since the number of samples in Xpert were around 100, I would not strongly emphasize on % resistant status in the study, do authors also want to comment on true resistance prevalence for TB in their study?

9) I would suggest authors to rephrase their conclusions about transmission since transmission studies highly depend upon various factors especially the amount of time spent in the social gatherings and daily activities due to which the extent of exposure in different areas might differ. Author should include more references which talk about the transmission risk factors now a days.

10) I would agree upon the statistics of true prevalence provided they will shed light whether the individuals were true TB patients, if they have been followed up. Also, it is unreasonable to compare the % prevalence of subdistricts where the number of subjects is different. For example, Fulpur and Sirajganj Sadar. It is highly likely to differ in that case. Is there any benefit of calculating the true prevalence in three districts as a whole because I see the number of subjects are almost similar in all three districts?

11) I would like to know the views of authors if they have not chosen diffuse prior method or have some prior information about prevalence, how would it affect their study? Can they quote some example or references?

6. PLOS authors have the option to publish the peer review history of their article (what does this mean?). If published, this will include your full peer review and any attached files.

Reviewer #1: No

Reviewer #2: No

Reviewer #3: **Yes: **Vartika Sharma

---

## [Author Response · Author response to Decision Letter 0]

30 May 2022

We thank all reviewers for their valuable comments to improve our manuscript. We used blue font for the changes we made based on the reviewers’ comments.

Reviewer #1: This Manuscript is technically sound and easy to understand with enough supporting data. However, this type of studies was carried out previously in Bangladesh. It was not clear why DOTS data is critical in compare to the previous studies. DOTS centers have been widespread since 2007, authors did not identified/clarified about whether sample population from previous studies visited DOT center or not. 

Response: Thank you. There are previously published studies on tuberculosis in Bangladesh. However, these are mostly based on the general population, or people in rural areas or slum dwellers. No study has reported true prevalence and identified risk factors among suspects attending DOTS in Bangladesh. This has now been clarified in the revised manuscript. Lines: 101-102.

Furthermore, authors highlighted MDR MTB in the conclusion with limited number of GeneXPert data due to financial limitation. It is suggested not to highlight the percentage of the MDR in the conclusion.

Response: Thank you. We have modified this sentence in the conclusion.

Reviewer #2: The authors in this manuscript investigated the true prevalence of TB among TB suspects and identified their risk factors and clinical symptoms. They also studied the rifampin sensitivity in those selected populations. The entire study was conducted in Bangladesh which has high prevalence of TB. The manuscript is well written and the findings described here would be of interest for better surveillance management of TB surveillance.

LIne 221-227; 296-298: What is the socio-economic difference between Phulpur and Muktagacha DOTS centers? Was there any previous report/observation of high and detection level between these two sites?

Response: Thank you. Socio-economic conditions of Phulpur and Muktagacha are similar but the total number of samples vary between these two DOTS centers. There is no previous report of high prevalence or detection between these two sites.

Line 308: Is there any other studies on TB determining the true prevalence in other parts of the world? Moreover, based of socio-economic difference of the major cities, urban and rural areas, how did the authors claim that TB suspects can be representative of the whole country?

Response: We did not find any published study reporting true human tuberculosis prevalence in other parts of the world. As DOTS centers are uniformly located in the country including cities, urban and rural areas we think these prevalence and risk factors estimates likely are representative of the whole country. We have now stated this in lines 321-324.

Line 309: Is there any previous report of Rif resistant TB in Bangladesh? If yes, please provide reference and compare your findings.

Response: Thank you. According to the global tuberculosis report 2019, the incidence of MDR/RR-TB was 3.7 per 100,000 populations in 2018 in Bangladesh. According to the global tuberculosis report 2020, the incidence of MDR/RR-TB was 2.0 per 100,000 populations in 2019 in Bangladesh (WHO, 2018, 2019). The denominator of these estimates were the general population however in this report, the denominator was TB cases. This is the reason for the relatively higher MDR-TB estimate than in previous reports. We have discussed this in lines 326-331.

Line 315: Please provide journal reference for microscopy dependent TB detection sensitivity.

Response: We have mentioned the sensitivity and specificity of the microscopy dependent TB detection �Gizaw et al. (2020)�at lines 155-157.

Reviewer #3: The manuscript entitled “Hierarchical true prevalence, risk factors and clinical symptoms of tuberculosis among suspects in Bangladesh” has been read carefully. Authors here would like to emphasize on the scenario of true prevalence of tuberculosis around suspected areas of Bangladesh. Their aim to provide information on true prevalence and risk factors contributing to TB transmission is encouraging. I would like to provide my concerns listed below which could enhance their manuscript for acceptance:

1) The foremost concern of this study is their assumption for TB detection based on a less sensitive cum specific technique called auramine staining and they mentioned it that its not possible to test all samples by Xpert method. I understand this limitation, but specificity is a major concern from sputum by using Auramine staining. NTM’s could also get detected leading to misdiagnose if the staining is not supported with some better culture techniques for a greater number of samples.

Response: Thank you for this comment. Auramine staining is the widely used technique for the detection of Tb in the DOTS center based surveillance system in developing countries. It is non-invasive and an easy to use technique with low sensitivity but high specificity (~99%) [Gizaw et al. 2020] and hence very low probability of false positive diagnosis. The probability of detecting NTM is also very low as high risk people [with clinical symptoms] only visit DOTS centers. 

2) I would not like the statement (line 26 in abstract) in Bangladesh until there are more subjects under investigation. Using province or some other word would better justify their mini study.

Response: We have added in the study areas of Bangladeshatline 26 of the abstract.

3) There are only two criteria to place suspected individuals in TB category one is sputum smear which is a poor technique. Saying that culture takes longer time for results but is a gold standard is not enough, since this study started 2 years back, authors may have gotten enough time to screen samples through culture to better understand this criterion.

Response: Auramine staining and LED microscopy is used in every DOTS centre in Bangladesh to detect acid fast bacilli, but facilities for sputum culture are available only in a few centers in the country.

4) Are there any false positives in their study design with the staining process, considering the specificity is only 68 %?

Response: The specificity of auramine staining is around 99%. So, there might be more false negatives and very few false positives [the sensitivity= 62.7% and the specificity is 98.7% (Gizaw et al. 2020)].

5) In selecting their divisions from specific areas of Bangladesh, they use random process, do they mean there is no prior information about number of people visiting, having likely TB symptoms?

Response: Yes, we did not know about the number of people visiting DOTS centers and having likely Tb symptoms before starting this study.

6) Similarly, for concluding risk factors for mixed effects multivariable logistics regression model, is there any prior value for the estimation of ODDS?

Response: We used non-informative prior for the estimation of the odds ratio.

7) The Odds ratios are impressive for the risk factors in univariable and multivariable logistic regression Bayesian model, I would like to know why only two risk factors are retained in the final model, is it because of the insignificant odds? It would have been interesting to understand how inclusion of smoking would affect the odds.

Response: Thank you very much for this comment. Yes, only two risk factors had significant odds ratios [95% confidence interval of the ratio excludes one]. If we add smoking in the model then the odds ratio for the age group ≤ 25 increase 21.7% which is below the threshold (25%) to consider smoking as a confounder. The odds ratio for other age categories and presence of tuberculosis patients in the family or neighborhood increase by <10%. We have added supplementary file 7 to show the changes in odds ratios after adding smoking to the final model. We have also discussed this in lines 285-289.

8) Since the number of samples in Xpert were around 100, I would not strongly emphasize on % resistant status in the study, do authors also want to comment on true resistance prevalence for TB in their study?

Response: We have modified this statement. Line, 326-331. We don’t have data to allow a comment on the true resistance prevalence for TB in Bangladesh to be made.

9) I would suggest authors to rephrase their conclusions about transmission since transmission studies highly depend upon various factors especially the amount of time spent in the social gatherings and daily activities due to which the extent of exposure in different areas might differ. Author should include more references which talk about the transmission risk factors now a days.

Response: We have rephrased the conclusion as suggested by the reviewer. We have included another reference about TB risk factors. Lines: 290-298. 

10) I would agree upon the statistics of true prevalence provided they will shed light whether the individuals were true TB patients, if they have been followed up. Also, it is unreasonable to compare the % prevalence of subdistricts where the number of subjects is different. For example, Fulpur and Sirajganj Sadar. It is highly likely to differ in that case. Is there any benefit of calculating the true prevalence in three districts as a whole because I see the number of subjects are almost similar in all three districts?

Response: The estimation of true prevalence from a follow-up study would be the ideal approach, but our study was a cross-sectional design. It is a well established approach to calculate the true prevalence considering the sensitivity and specificity of the diagnostic test used. We acknowledge that the numbers of samples from each subdistrict were not equal. However, even with an unequal number of subjects in the subdistricts, our purpose was to check any spatial clustering. We did not find any spatial clustering but some subdistricts had high prevalence. Comparison of prevalence estimates between subdistricts does not require a uniform sampling design; unequal sample sizes are reflected in the variance of the estimates.The overall prevalence (14.2%) estimated represents the overall prevalence in all districts.

11) I would like to know the views of authors if they have not chosen a diffuse prior method or have some prior information about prevalence, how would it affect their study? Can they quote some examples or references?

Response:

We used diffused prior information for the prevalence in the model to give more weight on the data rather than priors, because none of the previous studies included DOTS centers subjects in Bangladesh. Even if we use 20% prior prevalence in the model the posterior median prevalence does not change much [median prevalence 14.9%] (Supplementary file 8 added). We have discussed this and added one citation in the discussion section. Lines: 306-310.

---

## [Editor Report · Decision Letter 1]

28 Jun 2022

Hierarchical true prevalence, risk factors and clinical symptoms of tuberculosis among suspects in Bangladesh

PONE-D-22-00718R1

Dear Dr. Alam,

We’re pleased to inform you that your manuscript has been judged scientifically suitable for publication and will be formally accepted for publication once it meets all outstanding technical requirements.

Kind regards,

Pradeep Kumar, Ph.D.

Academic Editor

PLOS ONE
---

## [Editor Report · Acceptance letter]

4 Jul 2022

PONE-D-22-00718R1 

Hierarchical true prevalence, risk factors and clinical symptoms of tuberculosis among suspects in Bangladesh 

Dear Dr. Alam:

I'm pleased to inform you that your manuscript has been deemed suitable for publication in PLOS ONE. Congratulations! Your manuscript is now with our production department. 

Kind regards, 

on behalf of

Dr. Pradeep Kumar 

Academic Editor

PLOS ONE